# Anemochore Seeds Harbor Distinct Fungal and Bacterial Abundance, Composition, and Functional Profiles

**DOI:** 10.3390/jof8010089

**Published:** 2022-01-17

**Authors:** Dong Liu, Jie Cai, Huajie He, Shimei Yang, Caspar C. C. Chater, Fuqiang Yu

**Affiliations:** 1The Germplasm Bank of Wild Species, Yunnan Key Laboratory for Fungal Diversity and Green Development, Kunming Institute of Botany, Chinese Academy of Sciences, Kunming 650201, China; j.cai@mail.kib.ac.cn (J.C.); hehuajie@mail.kib.ac.cn (H.H.); sunshineyhj@126.com (S.Y.); 2Royal Botanic Gardens, Kew, Richmond TW9 3AE, UK; caspar.chater@gmail.com

**Keywords:** anemochore, light seed, endophytic microbiome, quantitative real-time PCR, MetaCyc genome database

## Abstract

Many plants adapted to harsh environments have evolved low seed mass (‘light seeds’) with specific dispersal strategies, primarily either by wind (anemochory) or water (hydrochory). However, the role of their seed microbiota in their survival, and their seed microbial abundance and structure, remain insufficiently studied. Herein, we studied the light seed microbiome of eight anemochores and two hydrochores (as controls) collected from four provinces in China, using qPCR and metagenomic sequencing targeting both bacteria and fungi. Substantial variations were found for seed endophytic fungi (9.9 × 10^10^~7.3 × 10^2^ gene copy numbers per seed) and bacteria (1.7 × 10^10^~8.0 × 10^6^). Seed microbial diversity and structure were mainly driven by the plant genotype (species), with weak influences from their host plant classification level or dispersal mode. Seed microbial composition differences were clear at the microbial phylum level, with dominant proportions (~75%) for Proteobacteria and Ascomycota. The light seeds studied harbored unique microbial signatures, sharing only two *Halomonas* amplicon sequence variants (ASVs) and two fungal ASVs affiliated to *Alternaria* and *Cladosporium.* A genome-level functional profile analysis revealed that seed bacterial microbiota were enriched in amino acid, nucleoside, and nucleotide biosynthesis, while in fungal communities the generation of precursor metabolites and respiration were more highly represented. Together, these novel insights provide a deeper understanding of highly diversified plant-specific light seed microbiota and ecological strategies for plants in harsh environments.

## 1. Introduction

Microbial influence is particularly important during the earliest phases of plant development, as seed germination and seedling growth are fragile life stages with major effects on plant populations and agricultural productivity. Understanding the determinants of microbiome composition during these early life stages is important for interpreting the entire process of microbiome assembly in plants and improving our ability to enhance plant health [1,2].

Seed production is a critical step in a plant’s life history. Seeds are a unique ontogenetic state, rich in starch, lipids, and proteins. A large number of studies have shown that plant seed microbial ecosystems are rich in microbial populations, which exist not only on the surface of seeds but also within embryos [3]. Previously, few studies on seed-borne bacteria have been published due to difficulties in culturing and isolation; however, with the development of high-throughput sequencing, many seed endophytic microbiota have been identified. For instance, bacterial genera *Pseudomonas*, *Sphingomonas*, *Pantoea*, and *Bacillus* have been shown to be abundant taxa in plants [4,5,6]. Similarly, a fungal genus *Alternaria* is also widespread within seed fungal microbiota, and many *Alternaria* species act as beneficial endophytes and can be utilized as biocontrol agents or sources of active compounds [7,8]. Seed microbiomes have been received increasing attention due to their beneficial roles during seed germination and seeding growth, as well as their impact on plant microbiome assembly and fitness of their host plants [3,9,10,11].

The seed microbiome is closely associated with the host plant microbiome and the plant’s habitat. Seed microbiota, as well as plant microbiota, can be acquired simultaneously from the environment and the parent via vertical transmission, which ultimately contributes to seed microbiome of the next generation [12]. David studied the conservation and diversity of seed associated endophytes in *Zea* and found that major seed bacterial microbiota in the *Zea* wild ancestor persist in diverse domesticated maize [13] and that seed bacterial endophyte community composition changed with its host plant phylogeny [13]. As the seed microbiome is influenced by its host plant microbiota, it could be expected that diverse plant families with divergent seeds may possess divergent seed-borne microbial flora. Investigations on crop seed microbiota have shown that members of seed bacterial microbiota are relatively conservative, as is their vertical transmission [14,15,16]. For these reasons, variations of seed bacterial microbiota may be greater at the plant family level compared to within genera or within species. In contrast, the mycobiome is a more flexible fraction within the seed microbiome and carries more environmental signatures, being mainly shaped by soil and host plant rhizosphere environments [6,17]. Therefore, the seed mycobiome fluctuates in response to surrounding abiotic and biotic fluctuations. Plants within the same genera are therefore likely to possess more diverse seed mycobiomes as a product of their host plant’s different geographic locations.

Several studies of microbial structure and function have revealed that, across a range of host-associated microbial niches, microbial populations are based more on function than on taxonomic structure [18,19,20]. Seeds are rich in starch, lipids, and proteins, and seed endophytes that share similar functions may compete for resource utilization, assimilation, or biosynthesis. Hence, predicted functional profiles of seed microbiomes appear to be fairly accurate [21]. Nevertheless, plants have evolved divergent seed dispersal and propagation strategies, primarily by wind (anemochory) or by water (hydrochory) [22]. Seeds that share the same dispersal mechanism often resemble one another in terms of physical appearance and morphology. For example, anemochory favors low density and feathery traits—so-called ‘light seeds’. Do these convergent seed traits therefore confer convergent seed microbiomes? Is shared seed dispersal strategy reflected also in shared core microbes and similar functional profiles? As dispersal mode is the outcome of evolution by natural selection, this trait could be of great important for selection of seed microbiomes with beneficial roles.

To better comprehend the influence of seed host plant differences (classification at family, genus, and species (genotype) levels, versus dispersal modes) on the seed microbiome, we collected eight anemochores and two hydrochores (as controls) and employed high-throughput sequencing to investigate the diversity of seed endophytic bacterial and fungal microbiota. Furthermore, in order to obtain microbial absolute abundance information, a quantitative real-time PCR approach was performed to quantify gene copy numbers for both bacteria and fungi.

**Hypothesis** **1** **(H1).***It was hypothesized that similar dispersal modes would select for seed microbiomes that are similar in abundance, diversity, and structure*.

**Hypothesis** **2** **(H2).***Within the microbiome, core taxa can**drive community composition and function irrespective of their abundance [23,24]**, and we hypothesized that plants with shared dispersal modes, such as anemochores, would share many core microbes*.

**Hypothesis** **3** **(H3)**. *Lastly, we hypothesized that anemochore seeds would exhibit similar microbial functional profiles that would in turn enhance their survival*.

To ascertain these functional traits, putative microbiome metabolic profiles were predicted using a database of reference genomes [25,26].

## 2. Materials and Methods

### 2.1. Plant Seed Collection and Surface Sterilization

Because diverse plant life features (herb, vine, and shrub) with divergent seeds may harbor different seed-borne microbiota, we selected ten plants that have evolved low seed mass (‘light seeds’) with specific dispersal strategies, including two hydrochoric herbs (CS, *Euryale ferox*; PFC, *Nuphar pumila*), two anemochoric herbs (QM, *Dianthus superbus*; CJSZ, *Dianthus repens*), three anemochoric vines (DHQT, *Illigera grandiflora*; XNFCZ, *Combretum griffithii*; SFCZ, *Combretum wallichii*), and three tree species (QGLR, *Terminalia myriocarpa*; DLR, *Terminalia franchetii*; CZLR, *Terminalia franchetii*).

To cover plant genetic diversity, each seed accession was collected from one plant population that was harboring over 50 individuals. Seed accessions of ten wild plant species used in this study were obtained from the Germplasm Bank of Wild Species, Kunming Institute of Botany, Chinese Academy of Sciences (Appendix A; Figure 1). All seed accessions had been stored at −20 °C after dehydration to a moisture content in equilibrium with 15% relative humidity. Seeds had been collected from 10 localities from four provinces in China (Appendix A). For each accession, 50 seeds were surface sterilized before DNA extraction. Sterilization steps were: full immersion for 10 s in ethanol, then 2 min in bleach, and lastly in 70% ethanol for 2 min, according to Arnold et al. (2007) [27]. Seeds were then surface dried using sterile absorbent paper, individually transferred to 5 mL tubes, and then frozen at −20 °C prior to DNA extraction.

### 2.2. Microbial DNA Extraction, PNA Clamps, PCR Amplification, and High-Throughput Sequencing

Frozen-sterilized seeds were pulverized using a Mixer Mill (MM400, Retsch, Germany), and then total community DNA was extracted using the Power Soil DNA kit (12888, MoBio^®^, Carlsbad, CA, USA) or Qiagen DNeasy Plant Mini Kit (Qiagen, Redwood City, CA, USA). Illumina amplicon sequencing of V5–V7 (hypervariable region of the bacterial 16S rRNA) and ITS regions were performed using two primer pairs: 799F-1193R (for endophytic bacteria) [28] and ITS1F-ITS1R (for endophytic fungi) [29]. In order to improve the efficiency of the polymerase chain reaction (PCR) and inhibit the amplification of mitochondrial and plastid templates [30], two types of peptide nucleic acid (PNA) clamps were included in the PCR mix (mPNA and pPNA for blocking mitochondrial and plastid DNA, respectively). An amount of 50 μL PCR reaction mix contained 25 μL 2× PCR mix, 2 μL of each primer (5 μM), 2.5 μL mPNA (5 μM), 2.5 μL pPNA (5 μM), 2 ng template DNA, and 16 μL ddH_2_O. PCR thermal cycling conditions were set under the following conditions: 94 °C for 3 min (initial denaturation), 30 cycles of 15 s at 94 °C, 75 °C 10 s, 55 °C 10 s, 68 °C 30 s, and concluded with a final extension for 10 min at 72 °C. Amplicons were then purified with the Gel Extraction kit (OMEGA bio-tek, Doraville, GA, USA), and DNA concentrations were measured with the Nanodrop 2000 (Thermo Scientific, Wilmington, DE, USA). Purified amplicons were combined in equimolar concentrations and pair-end sequenced on the Illumina Miseq-PE250 platform (Personalbio^®^, Shanghai, China).

QIIME 2 (Quantitative Insights into Microbial Ecology 2 (QIIME 2)) was used for processing the obtained sequences. Base N, sequence lengths < 160 bp, sequences with a mismatched base number > 1 of the 5′ end primer, sequences with >8 identical consecutive bases, as well as chimeric sequences, were all removed. To avoid the over-estimation of microbial diversity, singletons were also deleted. Filtered sequences were then clustered into operational taxonomic units (OTUs) at a 97% similarity cutoff, via searching reads against the Greengenes (for 16S rRNA) [31] and UNITE (for ITS) database, respectively [32,33]. After deleting super-minor OTUs (abundance < 0.001%), the rest were grouped based on their assigned taxonomic levels. All sequence data have been deposited to the ENA Sequence Read Archive under accession number PRJNA774071.

### 2.3. Quantitative Real-Time PCR

To quantify gene copy numbers of seed endophytic bacteria and fungi, quantitative real-time PCR (qPCR) was conducted using the same primer pairs: 799F-1193R for bacteria (10 μM each; [28]) and ITS1F-ITS1R for fungi (10 μM each; [29]). The 30 μL qPCR reaction mix contained 15 μL 2× qPCR mix, 1 μL of each primer, 3 μL template DNA, and 10 μL ddH_2_O. To estimate seed endophytic microbial gene abundances, bacterial standard curves were generated with a 10-fold serial dilution of a plasmid template in which the target gene amplified from the sample had been ligated to the T-vector. Fluorescence intensities were detected in a Real-Time System Light Cycler 480 II, 384 (Roche, Basel, Switzerland) with the following cycling conditions: 95 °C for 5 min, 40 cycles of 15 s at 95 °C, and 60 °C 30 s. Each seed DNA (biological) replicate was subjected to three independent qPCR runs (technical replicates), and the final gene copy number (X_0_) was calculated from a linear equation: Ct = −K × logX_0_ + b. Ct is the number of amplification cycles when the fluorescence signal of the amplification product reaches the set threshold, K is the slope of the standard curve, and B is the intercept of the standard curve. The standard curve was generated from the cycle threshold value to the known number of copies in the standards.

### 2.4. Statistical Analysis

Seed endophytic microbial alpha-diversity was estimated using the Shannon diversity index [34]. One-way analysis of variance (ANOVA) followed by Tukey HSD was used to compare significant variation of the Shannon index among plant seeds. Seed microbiome beta-diversity (among-plant species difference) was evaluated by pairwise Bray–Curtis distances [35] and visualized using non-metric multidimensional scaling (NMDS) plots. Permutational multivariate analysis of variance (PERMANOVA) was applied to test the significant differences (overall and pairwise) among the plant seeds shown in NMDS plots. Microbial taxa differences among plant species were evaluated by core and unique microbe (CUM) and UPGMA (unweighted pair-group method with arithmetic mean) clustering analyses [36]. Specifically, CUM analysis was based on microbial ASV data (amplicon sequence variant; a term used to describe taxonomy based on DNA sequence similarity) and visualized using a petal diagram. UPGMA clustering was performed according to the Pearson correlation coefficient matrix of seed microbial composition data (Euclidean distance-based) and arranged according to sample clustering results.

Seed bacterial and fungal marker-gene (16S rRNA and ITS) amplicon sequences were used for functional prediction based on the MetaCyc database incorporated in PICRUSt2 (Phylogenetic Investigation of Communities by Reconstruction of Unobserved States 2) software (detailed in https://github.com/picrust/picrust2/wiki, accessed on 20 February 2021). MetaCyc contains pathways (PWY) involved in primary and secondary metabolism (level 2), as well as many functional units including metabolites, reactions, enzymes, and genes [37]. PCoA (principal co-ordinates analysis) was used to capture seed microbiome functional similarity by reducing the complexity of functional unit data from PICRUSt2 analysis.

## 3. Results

### 3.1. Bacterial and Fungal Abundances in Wind- or Water-Dispersed Seeds

Significant differences in microbial abundances were observed among the ten light seeds from anemochoric plants, with the highest values observed in CZLR (Figure 1). For seed endophytic bacteria, 16S rRNA gene copies ranged from 1.7 × 10^10^ in CZLR to 8.0 × 10^6^ in SFCZ (over 2000-fold difference). For seed endophytic fungi, ITS gene abundance ranged from 9.9 × 10^10^ in CZLR to 7.3 × 10^2^ in CS (1.3 × 10^7^–fold difference). For individual seed microbiomes, bacterial abundances were consistently higher than those of fungi, ranging from 1.5 (in QGLR) to 31,711 times (in CS) (Appendix A).

### 3.2. The Composition of Bacterial and Fungal Taxa within Wind- or Water-Dispersed Seeds

After quality-filtering and removing singleton and chimeric sequences, the clear high-quality 16S rRNA reads (length of 371–381 bp) obtained from seeds ranged from 56,379 to 132,631 per sample (mean = 103,589) and assigned to a total of 48,681 ASVs.

The filtered clear ITS dataset reads (length of 185–364 bp) ranged from 43,077 to 125,231 per sample (mean = 105,696), belonging to 3767 ASVs. ASV tables were rarefied to 4506 bacterial and 3679 fungal sequences per sample, according to the samples with minimum sequences.

The major endophytic bacterial taxa across all seeds belonged to Proteobacteria (average 78%) and Actinobacteria (16%). These two groups were responsible for more than 90% of the total bacterial sequences obtained. Groups of Firmicutes, Bacteroidetes, Chloroflexi, Armatimonadetes, Fusobacteria, Acidobacteria, and Deinococcus-Thermus were less abundant (relative abundance < 1%) but were still identified in all seeds. Substantial changes were observed at the phylum level. The relative abundance of Proteobacteria ranged from 37% (in DLR) to 96% (in CS), and Actinobacteria varied from 2% (in QM) to 37% (in DLR) (Figure 2A). At the genus level, the mean relative abundance of *Halomonas* (38%) was the highest, followed by *Pseudomonas* (3.9%), *Curtobacterium* (3.2%), *Nesterenkonia* (2.3%), *Sphingomonas* (2.3%), *Aureimonas* (2.2%), *Chryseobacterium* (2.1%), *Pantoea* (2.1%), *Methylobacterium* (1.5%), and *Massilia* (1.2%) (Figure 2C). Among them, only the most highly abundant genus *Halomonas* was shared among all the seeds, but there were broad variations in their relative abundance, from 0.3% in CZLR to 88% in SFCZ.

Seeds belonging to Nymphaeaceae (CS, PFC) species were found to share a similar seed endophytic bacterial composition, with *Halomonas* dominant (~70%). The composition of endophytic bacterial communities in the seeds of Caryophyllaceae (QM, CJSZ) species was significantly different, dominated by *Pseudomonas* (25.6%) and *Pantoea* (18%), respectively. The five Combretaceae species represent two genera. XNFCZ and SFZC, belonging to *Combretum*, showed substantial variation in their bacterial composition: XNFCZ seed bacterial microbiota were mainly occupied by *Aureimonas* (20%), *Sphingomonas* (13%), and *Methylobacterium* (7%), while SFCZ was predominated by *Halomonas* (88%). Unique seed bacterial compositions were also found for QGLR, DLR, and CZLR, all of which are in the genus *Terminalia* (Figure 1).

The main fungal phyla in the selected species’ seeds were Ascomycota (75%) and Basidiomycota (9.5%), followed by Mucoromycota (0.1%) and Olpidiomycota (0.06%). The relative abundance of Ascomycota ranged from 40% (in DHQT) to 95% (in CJSZ and CZLR), and Actinobacteria varied from 2% (in QM) to 37% (in DLR). The relative abundance of Ascomycota ranged from 0.2% (in CJSZ) to 28% (in DLR) (Figure 2C). For PFC seeds, the majority (98%) of fungal sequences could not be clustered into definite phyla and instead included unidentified taxa and unclassified fungi.

Endophytic fungal communities were conspicuously occupied by distinct fungal genera among the seeds. Only three fungal genera were represented across all seeds, and their relative abundance varied greatly: *Fusarium* (from 0.02% in CJSZ to 52% in CZLR), *Cladosporium* (from 0.1% in XNFCZ to 3.5% in QM), and *Epicoccum* (from 0.02% in PFC to 8% in CJSZ) (Figure 2D).

### 3.3. Identification of the Main Drivers of Microbial Diversity and Structure within Wind- and Water-Dispersed Seeds

The endophytic bacterial and fungal diversity of the seeds was assessed using the Shannon diversity index, and significant differences (*p* < 0.05) between each grouping category (plant species, genus, and family) were calculated using Tukey’s HSD pairwise comparisons at the saturated rarefaction depth.

When the seeds were grouped by higher taxonomic ranks (either at family or genus level), no significant differences in endophytic diversity (for either fungi or bacteria) were found. At the plant species level, however, seed bacterial diversity index ranged from CZLR’s high 7.9 to SFCZ seeds’ low 2.2; CZLR also scored highest for seed fungal diversity index (5.0), whereas PFC seeds scored the lowest (1.0) (Table 1). The diversity of seed bacteria was generally higher than that of fungi, with the exception of CS, SFCZ, and QFLR (Table 1). At the plant species level, there were 38 and 20 plant seed pairs with significantly different seed endophytic bacterial and fungal diversity indices, respectively (Table 1). This suggests that light seed (anemochory/hydrochory) bacterial diversity is therefore more dependent on the plant species than is fungal diversity.

In order to evaluate the major drivers of seed microbial community composition, beta-diversity analysis was conducted by non-metric multidimensional analyses (NMDS) (Figure 3) in combination with PERMANOVA test. There was strong variation in seed microbiomes. Seed bacterial community structure between plants of the same genus can vary substantially (e.g., *Terminalia* species DLR, QGLR, CZLR). Likewise, plants from different genera (such as CS, PFC, CFZC, QGLR) can share similar seed bacterial microbiota, as indicated by the clustering together of their community structure NMDS coordinate points (Figure 3). For fungi, PFC and DHQT seed fungal communities showed the largest intra-group variation (as indicated by the shape of the ellipse) and obviously differed from other seed fungal communities (Figure 3).

Among the grouping factors, plant species or ‘genotype’ was observed as the main driver of seed microbiome composition, which was the case for both bacteria (PERMANOVA test, sample size = 30, *F* = 3.31, *p* = 0.001) and fungi (PERMANOVA test, sample size = 27, *F* = 4.83, *p* = 0.001). These results indicate the strong variation in seed microbiome structure, which is more dependent on plant species.

### 3.4. Species Differences and Marker Species Analysis

Having explored differences in the composition of microbial communities (beta diversity), we sought to understand which species were responsible for these differences.

Only two bacterial and fungal ASVs were shared across the seed microbiomes of all ten different species (Figure 4). ASV_9142 and ASV_2289 were assigned to an unclassified *Halomonas* (Appendix A). ASV_1637 and ASV_1889 were assigned to an unclassified *Alternaria* and unclassified *Cladosporium* (Appendix A).

There were strong differences in the number of unique ASVs within the anemochores, ranging from 25 (in SFCZ) to 725 (in CZLR) for bacteria (Figure 4A) and from 40 (in SFCZ) to 320 (in CZLR) for fungi (Figure 4B). Among anemochores, the average unique bacterial ASVs (270) were approximately 2 times higher than that of the fungal ASVs (155).

Compared with the control (two hydrochores: CS and PFS), the average number of total microbes was around 2 times higher in anemochores (424) than hydrochores (244). Hydrochore seeds also harbored more unique fungal ASVs (250) than bacterial ASVs (averaged 60).

Two-way clustering of the relative abundance of the top 10 microbial genera showed no clear separation of the samples by factors of dispersal mode or seed taxonomic rank at either the family or genus level (Figure 4). Seeds of different species accumulate their unique microbial signatures. For bacterial genera, *Methylobacterium* and *Sphingomonas* were very abundant in XNFCZ seeds but scarce in others. *Pantoea* and *Pseudomonas* were abundant in CJSZ and QM, respectively (Figure 4C). For fungi, commonly studied genera such as *Alternaria*, *Didymella*, and *Fusarium* were highly abundant in seeds of CJSZ, XNFCZ, and CZLR, respectively (Figure 4D). Seeds of all species tend to be selectively abundant in one or two diverse microbial genera.

### 3.5. Microbiome Function

Using data analysis methodologies, we can predict the composition of seed microbial community genes or functional units by referring to known microbial genome data for samples, using only the sequence data of microbial community marker genes (such as 16S rRNA and ITS used here). Based on this, we can infer an overview of the functional potential of the seed microbiome in tested samples and can maximize the cost-effective advantages of amplicon sequencing.

The function here refers to gene families, such as KEGG homologous genes (KO), EC enzyme classification numbers, COG, etc. Since the number of functional units (EC/KO/COG) is often too large to be directly compared, we can also use the sample difference distance matrix (Bray–Curtis distance) and principal coordinate analysis to expand sample function difference in a lower dimension.

Functional unit differences in seed bacterial communities were mainly explained by PcoA 1 (56.2%), which was consistent with the distribution of unique seed microbial ASV numbers. The potential function of the seed bacterial community with low (<100, four left-distributed plants), medium (three central-distributed plants), and high (>300, four right-side plants) unique microbial number showed distinct functional profiles along PcoA 1 (Figure 5A). For fungi, PCoA 1 explained 74.7% of the functional variation (Figure 5B). Similar to what we report for seed entophytic fungal community structure (Figure 3), PFC exhibited substantial difference (distributed at the very left corner) in seed fungal profiles compared with other species. DHQT showed the largest intra-group variation (as indicated by the shape of the ellipse) in both fungal community structure (Figure 3) and functional profile (Figure 5B).

As each species’ seeds were shown to possess a unique microbial functional profile, we therefore clustered all selected plant species and examined the main functional types identified in their seed endophytic bacterial and fungal microbiota (Figure 6). Using functional abundance prediction based on the genome database, it was inferred that seed bacterial microbiota have stronger metabolic functions than fungi. There were seven main functional types (degradation, generation of precursor metabolite and energy, macromolecule modification, metabolic clusters, and detoxification) constituted by 60 functional pathways (Appendix A). Among these specific functional types, there were 58 significantly different pathways, except for pyrimidine deoxyribonucleotide biosynthesis from CTP and pyrimidine deoxyribonucleotide de novo biosynthesis IV (Appendix A). Biosynthesis of amino acids, cofactors, prosthetic groups, carrier, vitamins, nucleosides, and nucleotides were the strongest metabolic types inferred for seed bacterial microbiota (Figure 6A). For the seed endophytic fungal community, nucleoside and nucleotide biosynthesis, electron transfer, and respiration were highly enriched (Figure 6B). Other metabolic functions such as fatty acid and lipid degradation, glycan biosynthesis, and tRNA charging were also highly abundant (Figure 6B).

## 4. Discussion

### 4.1. Changes in the Seed Microbiome among Anemochores

We initially hypothesized (H1) that due to similar dispersal modes among the species, their seed microbiomes would share similar microbial abundance. However, in contrast with H1, the seed microbiomes studied showed a high degree of plant species specificity, and seed endophytic microbial diversity and structure were mainly driven by plant species or ‘genotype’. Seed microbiome differences were reflected in aspects of bacterial and fungal abundance, composition, diversity, and structure. Based on qPCR analysis, we found significantly different absolute abundances of endophytic bacteria and fungi in the seeds of eight anemochore species, and the variation in fungi (maximum 1.3 × 10^7^ times) was greater than that of bacteria (maximum 2.1 × 10^3^ times). Greater variation in fungal abundance could be a reflection of their diverse habitats, as seed endophytic fungi are classified as a more variable fraction and more easily influenced by fluctuations in their host plants’ environments [17].

As well as orders of magnitude variations in seed microbial abundance in bacterial and fungal abundances among plant species, there were also significant differences in the ratios of bacteria to fungi (B/F), as evaluated by 16S/ITS copies. For example, anemochores from same plant family (Combretaceae) possessed seed B/F ratios ranging from 1.5 to 8361. For the two hydrochores, the seed B/F ratio was even larger (61 in PFC vs. 31,711 in CS). Such extreme variation could be attributed to differences in plant species’ habitats because seeds of alpine plants grown in the same soil and environmental conditions have a consistent B/F ratio (~1) [5]. On the other hand, the seeds themselves vary in their size, fruit morphology, chemical composition, and anatomic features. Within the limited variables tested here, our results suggest that plant species, or genotype, could be a driving force of seed endophytic diversity. Furthermore, seed bacterial diversity was generally higher than fungal diversity. In contrast with the findings for alpine plants, our results indicate that the light seed (anemochore/hydrochore) bacterial microbiome has a higher plant species specificity than the mycobiome. Seed bacterial microbiota appear to be relatively conservative and, in most cases, vertically transmitted from their host plant [14,15,16], while the seed mycobiome is more diverse and mainly shaped by soil and host plant rhizosphere conditions [17].

From the perspective of changes in seed endophytic microbial diversity and structure, plant dispersal mode may be an insignificant factor in shaping the seed microbiome. Seed microbiome was mainly dependent on the plant species. We hypothesized that due to the same dispersal mode, anemochores might share plenty of the same microbes (H2). However, we found that seed microbiomes were dissimilar despite shared taxonomic ranks at both the family and genus level. Our results show that plants from the same genus can exhibit differences in (i) the total number of unique seed microbial ASVs (i.e., 319 vs. 1045 between *Terminalia* species) and (ii) the ratio of bacterial to fungal unique seed microbial ASVs (0.12 vs. 4.4 within the *Terminalia* species).

Each plant species studied has its unique microbial ‘fingerprint’, but at the microbial genus level, they have common features. Some seed endophytic bacterial genera known to be beneficial to host plants [37,38], such as *Sphingomonas*, *Pseudomonas*, *Halomonas*, and *Pantoea*, were also detected in all seeds. Some seed bacterial genera, such as *Sphingomonas* and *Pantoea*, are also present in seeds of alpine plants [5] and many crops [38]. Especially, bacterial endophytes such as *Pantoea*, *Microbacterium*, *Bacilus*, *Sphingomonas* have been found to be widespread across maize cultivars [13,39].

Moreover, we found highly abundant *Halomonas* in most anemochores. *Halomonas* belongs to Gram-positive bacteria known to have a strong tolerance for salt, pH variation, and temperature fluctuation [40]. The two *Halomonas* ASVs were identified as highly abundant core ASVs among all the seeds tested, suggesting that these stress-tolerant microbes may play a role in the survival of plants with light seeds.

Two seed endophytic fungal genera (*Alternaria* and *Didymella*) have been found to have plant-pathogenic potential [41,42], although all the seeds were collected from wild healthy plants. Diverse fungal species belong to *Alternaria* and *Didymella*; although using second-generation amplicon sequencing, the resolution of this information is still limited, and some fungal species may not always exhibit pathogenic potential. On the other hand, in contrast with the positively interacting bacterial network, the seed mycobiome tended to be mutually exclusive, which could show resilience towards plant pathogens [5]. Two core fungal ASVs affiliated to *Alternaria* and *Cladosporium* are classified as dematiaceous fungi, capable of persisting in harsh ecological conditions and also widely distributed in soil and plants [6,43,44]. Some secondary metabolites of *Alternaria* have insecticidal, bactericidal, and antiprotozoan activities [45]. The above-mentioned features may help explain why these core taxa can be shared among different plants and transmitted from generation to generation. In future studies, based on high-throughput culture methods, core taxa could be isolated and reconstructed as simple synthetic communities, to use them and identify their beneficial effects to host plant species and their transmission to the next generation of seeds under different abiotic and biotic stress conditions.

### 4.2. Predictive Functional Profiling in the Seed Microbiome

In contradiction to our hypothesis (H3), anemochore seed microbial functional profiles showed clear differences among plant species, and this change was similar with variations in community structure. There can be a disconnect between microbial community structure and function in soil or forest ecosystems due to differences between the drivers of microbial growth and those of microbial function [46,47]. However, we found that within the seed micro-habitat, both endophytic microbial structure and function were mainly driven by plant species.

Among different plant species with light seeds, PICRUSt functional prediction related to biosynthesis was highly represented in the seed microbiome. Seed bacterial microbiota were also significantly involved in amino acid, nucleoside, and nucleotide biosynthesis; their overrepresentation in bacterial communities indicates their capacities for secondary metabolism gene repository in the seeds. Nucleotides, along with nucleic acids, are distributed in the nuclei and cytoplasm of various organs, tissues, and cells in seeds, and the enhanced biosynthesis of these compounds may demonstrate the seed microbiome’s active roles in development and growth [46,47]. Results from the prediction of the seed bacterial microbiota of the medical plant *Salvia miltiorrhiza* also suggested that key bacterial communities may play active roles in the biosynthesis of specific pharmacologically active biomolecules [6]. Although this is predictive, these results are indicators that the seed bacterial microbiota are a rich reservoir for secondary metabolism. In addition, we found that predicted metabolic processes resulting in electron transfer and respiration were also significantly higher in seed fungal communities than in the bacterial communities, which might indicate a higher generation of precursor metabolites and energy by the seed mycobiome.

Although the ecological impacts of observed anemochore seeds have not been experimentally tested, most anemochore seed endophytes such as *Methylobacteria*, *Pantoea*, and *Pseudomonas* were also identified in other plants (maize cultivars). They exhibit the ability to stimulate plant growth by phosphate solubilization and pathogen antagonization [13]. An application of fungicides with the *Bacillus subtilis* strain was identified as a promising strategy for controlling *Fusarium verticillioides*-caused diseases [48]. Moreover, it is commonly accepted that seed endophytes could spread systemically through the seeding phyllosphere plant, exit the root, and colonize the rhizosphere [13,21], playing vital roles in plant health, disease resistance, and growth-promotion [13,49]. For anemochore, the extensive existence and co-evolution of these seed endophytes with their host plant might be a key biotic factor for plant survival in harsh environments.

## 5. Conclusions

Light seed microbiomes have not previously been paid special attention. However, their shared unique biotic features may help reveal traits that support the mechanism of long-distance dispersal and survival for plants growing under harsh conditions. We expected light seeds might share similar endophytic microbial communities, and this work sought to identify core taxa. However, plant species with light seed harbor their unique microbial ‘fingerprint’, either in terms of absolute abundance, microbial diversity, or in composition, structure, and functional profile. Based on our results, it was shown that plant dispersal mode is an exterior feature but arguably not the main driver for light seed microbiomes. For future studies, specific focus on the influence of plant species and genotype is recommended, as these appear to be the stronger drivers shaping the seed microbiome. Moreover, in terms of utilizing microbiomes, isolating and high-throughput culturing methods could be beneficial for constructing a simple synthetic microbial community to identify their accommodation to the seed micro-habitat and the influence of changes in the external environment.

## Figures and Tables

**Figure 1 jof-08-00089-f001:**
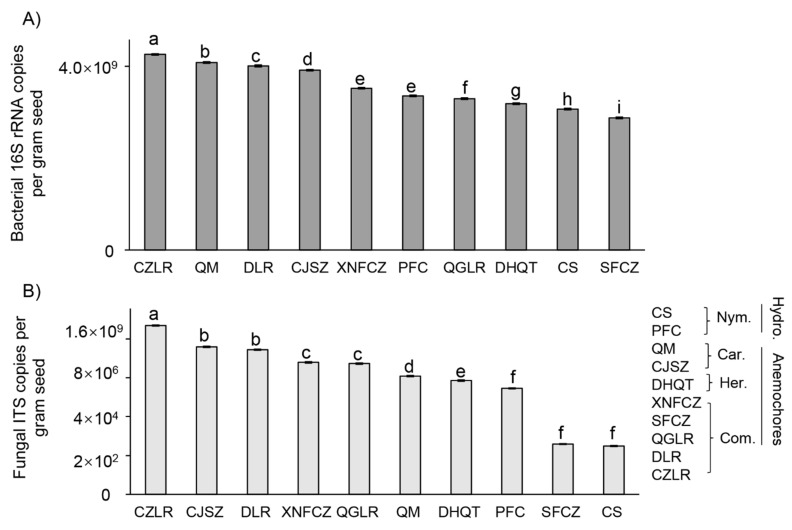
Seed microbial gene copy numbers determined by qPCR. Values are given by primers targeting (**A**) bacterial 16S rRNA and (**B**) the fungal ITS region in the seeds of two hydrochoric herbs (CS, *Euryale ferox*; PFC, *Nuphar pumila*) and eight anemochores including two herb species (QM, *Dianthus superbus*; CJSZ, *Dianthus repens*), three vine species (DHQT, *Illigera grandiflora*; XNFCZ, *Combretum griffithii*; SFCZ, *Combretum wallichii*), and three tree species (QGLR, *Terminalia myriocarpa*; DLR, *Terminalia franchetii*; CZLR, *Terminalia franchetii*). Lowercase letters above the bars indicate significant differences (*p* < 0.05, ANOVA, Tukey HSD) of gene copies and the *y*-axis represents the unit on a log 10 scale. The abbreviations behind the bracket indicate plant family name (Car, Caryophyllaceae; Com, Combretaceae; Her, Hernandiaceae; Nym, Nymphaeaceae) and their assigned dispersal mode (hydrochores/anemochores). Detailed seed host plant pictures are presented in Appendix A.

**Figure 2 jof-08-00089-f002:**
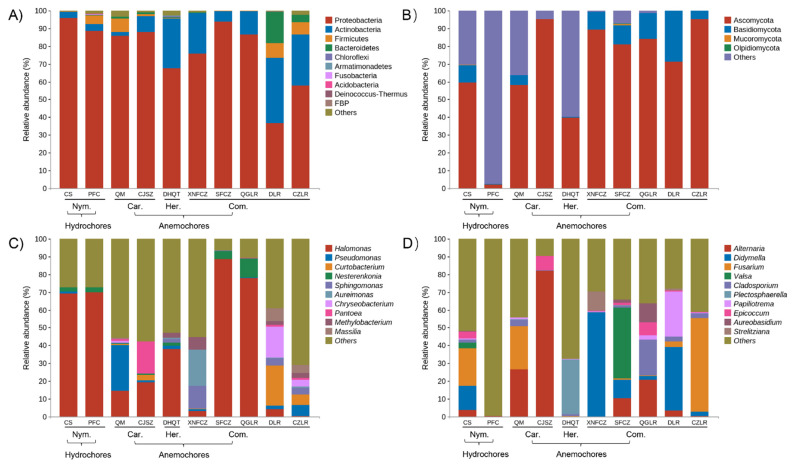
Abundant endophytic bacterial (**A**,**C**) and fungal (**B**,**D**) taxa found in the seeds of eight anemochores and two hydrochores. Relative abundance of uncultured and unclassified/unidentified taxa are grouped as ‘Others’. Values are the mean of three replicates. Full names of species abbreviations are described in Figure 1, and the abbreviations below the underline indicate the plant family name (Car, Caryophyllaceae; Com, Combretaceae; Her, Hernandiaceae; Nym, Nymphaeaceae) and their assigned dispersal mode (hydrochores/anemochores).

**Figure 3 jof-08-00089-f003:**
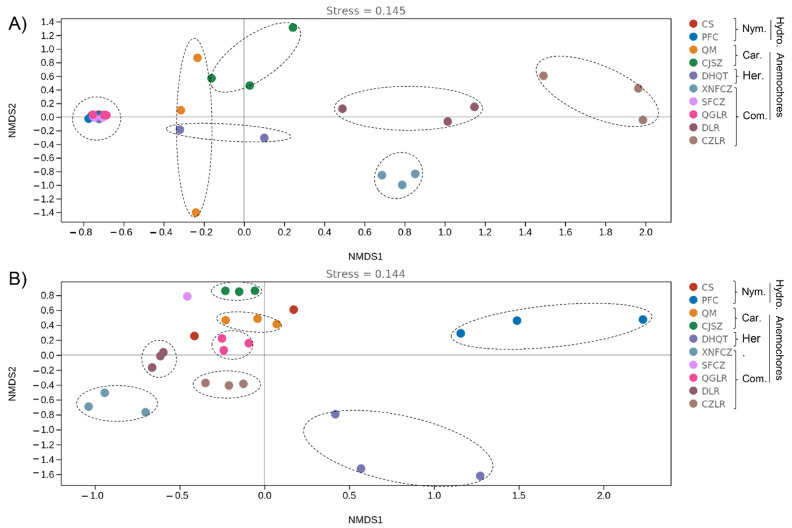
Seed endophytic bacterial (**A**) and fungal (**B**) community compositions as indicated by non-metric multidimensional scaling plots (NMDS) of pairwise Bray–Curtis distance in the eight anemochores and two hydrochores. The abbreviations behind the bracket indicate plant family name (Car, Caryophyllaceae; Com, Combretaceae; Her, Hernandiaceae; Nym, Nymphaeaceae) and their assigned dispersal mode (hydrochores/anemochores), including two hydrochoric herbs (CS, *Euryale ferox*; PFC, *Nuphar pumila*), two anemochoric herbs (QM, *Dianthus superbus*; CJSZ, *Dianthus repens*), three anemochoric vines (DHQT, *Illigera grandiflora*; XNFCZ, *Combretum griffithii*; SFCZ, *Combretum wallichii*), and three tree species (QGLR, *Terminalia myriocarpa*; DLR, *Terminalia franchetii*; CZLR, *Terminalia franchetii*).

**Figure 4 jof-08-00089-f004:**
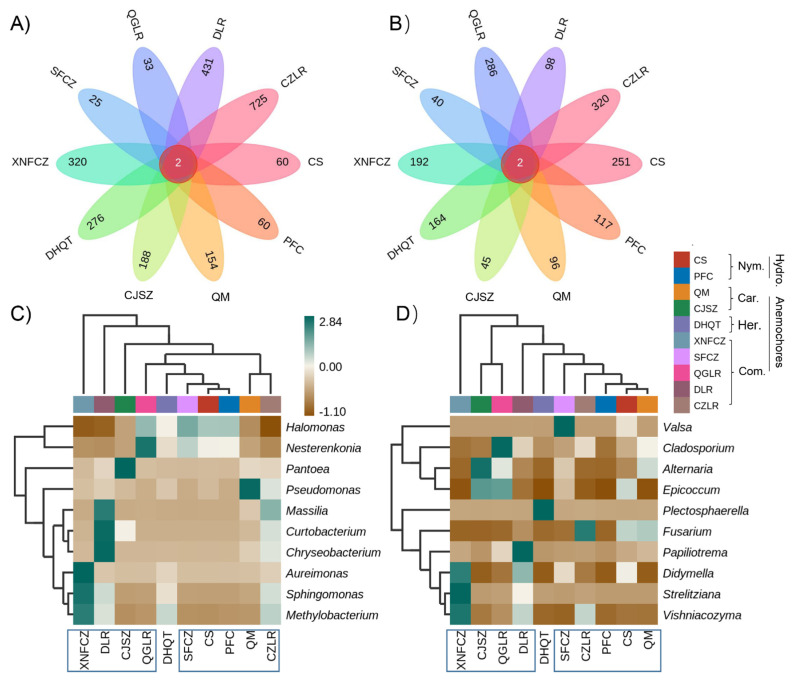
Petal diagram of the numbers of core and unique bacterial (**A**) and fungal (**B**) amplicon sequence variants (ASVs) and the double cluster heatmaps of the relative abundance of the top 10 bacterial (**C**) and fungal (**D**) genera within seeds of eight anemochores and two hydrochores. The horizontal and vertical clusters represent the groupings based on plant species and seed endophytes, respectively. Heatmap colors represent the averaged relative percentage of bacterial/fungal genera within each plant seed (n = 3). Color squares shifted towards green indicate higher abundance. The abbreviations behind the bracket indicate plant family name (Car, Caryophyllaceae; Com, Combretaceae; Her, Hernandiaceae; Nym, Nymphaeaceae) and their assigned dispersal mode (hydrochores/anemochores), including two hydrochoric herbs (CS, *Euryale ferox*; PFC, *Nuphar pumila*), two anemochoric herbs (QM, *Dianthus superbus*; CJSZ, *Dianthus repens*), three anemochoric vines (DHQT, *Illigera grandiflora*; XNFCZ, *Combretum griffithii*; SFCZ, *Combretum wallichii*), and three tree species (QGLR, *Terminalia myriocarpa*; DLR, *Terminalia franchetii*; CZLR, *Terminalia franchetii*).

**Figure 5 jof-08-00089-f005:**
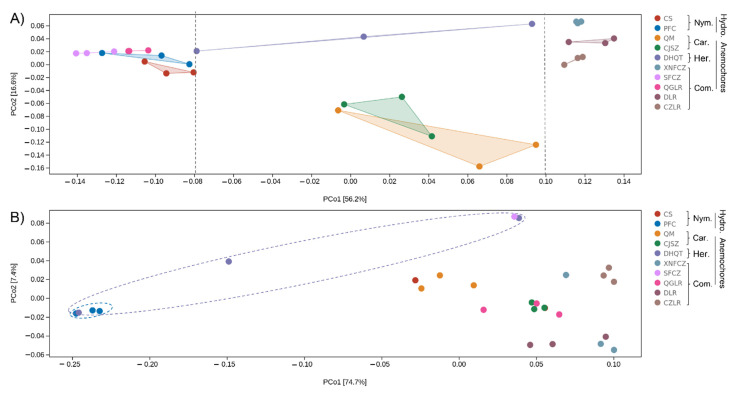
PCoA (principal co-ordinates analysis) of seed endophytic bacterial (**A**) and fungal (**B**) functional units (metabolism pathways and EC enzyme classification number from MetaCyc database). Dots of same color represent plant species. Percentages in brackets represent the proportion of sample variance data (Bray–Curtis distance matrix) that can be explained by the corresponding axes. The closer the distance of the two points on the coordinate axis, the more similar the functional composition of the two samples. The abbreviations behind the bracket indicate plant family name (Car, Caryophyllaceae; Com, Combretaceae; Her, Hernandiaceae; Nym, Nymphaeaceae) and their assigned dispersal mode (hydrochores/anemochores), including two hydrochoric herbs (CS, *Euryale ferox*; PFC, *Nuphar pumila*), two anemochoric herbs (QM, *Dianthus superbus*; CJSZ, *Dianthus repens*), three anemochoric vines (DHQT, *Illigera grandiflora*; XNFCZ, *Combretum griffithii*; SFCZ, *Combretum wallichii*), and three tree species (QGLR, *Terminalia myriocarpa*; DLR, *Terminalia franchetii*; CZLR, *Terminalia franchetii*).

**Figure 6 jof-08-00089-f006:**
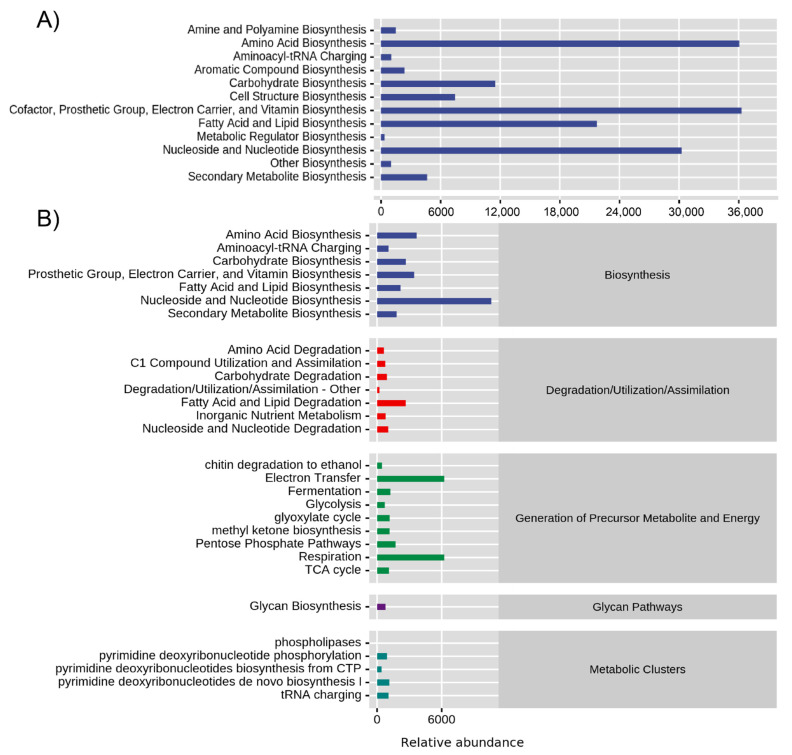
Seed endophytic bacterial (**A**) and fungal (**B**) community functional abundance prediction based on MetaCyc genome database. The horizontal axis is the relative abundance of the functional pathway (unit is the PWY (pathways involved in metabolism) per million) of the second classification level of MetaCyc, and to the right is described the first-level pathway to which this functional pathway belongs. Functional pathway refers to the average abundance of all samples, and detailed information is described in Figure 1.

**Table 1 jof-08-00089-t001:** Bacterial and fungal Shannon diversity indices in the seeds of eight anemochores and two hydrochores.

		CS	PFC	QM	CJSZ	DHQT	XNFCZ	SFCZ	QGLR	DLR	CZLR
Bacteria	Mean	3.5 D	3.2 D	5.0 C	6.0 B	5.2 C	6.1 B	2.2 E	2.6 D	6.6 B	7.9 A
	SD	0.3	0.4	0.7	0.2	1.0	0.0	0.2	0.2	0.4	0.2
Fungi	Mean	4.8	1.0 C	3.3 B	1.2 C	2.4 B	2.6 B	4.2	4.6 A	3.3 B	5.0 A
	SD	NA	0.2	0.2	0.4	0.9	1.5	NA	0.1	0.1	0.4

Values followed by different letters among rows indicate significant differences at *p* < 0.05 (ANOVA) between means (Tukey’s HSD pairwise comparisons, *n* = 3). NA: standard deviation (SD) was not captured due to limited replicates for PFC (two replicates) and SFCZ (one replicate). Plant abbreviations are described in Figure 1.

## Data Availability

Not applicable.

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
