# Peer review of "Anemochore Seeds Harbor Distinct Fungal and Bacterial Abundance, Composition, and Functional Profiles"

_jof, 2022, doi:10.3390/jof8010089_

Round 1

Reviewer 1 Report

This is a well done and very well presented work. I have no serious objections. Instead of loading entire file I am pointing at few language mistakes here. - please see lines:

16 - "bacteria" instead of "bacterial"

21- I would delete "respectively" - there are no two numbers to refer to this wording.

48 - "received" instead of "paid"

99- add "was" after "that"

435 - divide antiprotozoanactivity

Additionally, I would propose including more detailed interpretation of the ecological impact of the results.

Author Response

This is a well done and very well presented work. I have no serious objections. Instead of loading entire file I am pointing at few language mistakes here. - please see lines:

16 - "bacteria" instead of "bacterial"

Answer: Correction has been made.

21- I would delete "respectively" - there are no two numbers to refer to this wording.

    Answer: We have deleted “respectively”.

48 - "received" instead of "paid"

Answer: Correction has been made.

99- add "was" after "that"

Answer: Correction has been made.

435 - divide antiprotozoanactivity

Answer: Correction has been made.

Additionally, I would propose including more detailed interpretation of the ecological impact of the results.

Answer: Based on the reviewer’s valuable suggestion, we have included a paragraph in the discussion part to interpret the ecological impact of our results.

“Although the ecological impacts of observed anemochore seeds have not been experimentally tested, most anemochore seed endophytes such as Methylobacteria, Pantoea and Pseudomonas, were also identified in other plants (maize cultivars). They exhibit the ability of stimulating plant growth by phosphate solubilization and pathogen antagonization [13]. An application of fungicides with the Bacillus subtilis strain was identified as a promising strategy to control Fusarium verticillioides-caused diseases [49]. Moreover, it is commonly accepted that seed endophytes could spread systemically through the seeding phyllosphere plant, exit the root and colonize the rhizosphere [13,21], playing vital roles in plant health, disease resistance and growth-promotion [13,50]. For anemochore, the extensive existence and co-evolution of these seed endophytes with their host plant, might be a key biotic factor for plant surviving in harsh environments.”

Reviewer 2 Report

The present manuscript presents a study that starts from a strong hypothesis and works in a scientifically-sound way to investigate the proposed thematic.

The methods employed to investigate the microbiota are correct and coherent but, in the materials and methods section, giving some more explanation regarding the rationale behind the selection of the species to investigate would be helpful to the readers later on. Likewise, including some info, even just the list of species, in the main text and not only in the Supplementary Material would be appreciated.

The results section would need a great overhaul of the figures presented, which all lack in clarity that could help the readers.

Figure 1 is hard to compare due to the scale. Using a simple log10 scale on the Y-axis should, in my opinion, allow direct comparison of the quantities without relying on split panels and still not managing to understand what some bars represent (for example, SFCZ in panel A... it's below 4.00E7 but above 0... how much is it?).

On all the figures, using different colors or marker shapes to identify different plant families and - most importantly - to allow the readers to immediately identify the anemochores and the hydrochores would make the figures much, much easier to interprete for a reader. As it is, I had to continuously go back and forth between figures and the first caption to make sense of the pictures and see what the results were saying.

Lastly, while the discussion section is appropriate, I see that the authors did not cite many works regarding the seed microbiota of cereals.
Especially regarding the considerations of H1, that the plant-seed microbiota showed different, species- or even genotype-specific signatures, there are already indications of things going in that direction in the literature.
I therefore suggest one or more of the following articles for consideration in discussing the results and/or for the introduction:

Rijavec, T.; Lapanje, A.; Dermastia, M.; Rupnik, M. Isolation of bacterial endophytes from germinated maize kernels. Can. J. Microbiol. 2007 53, 802-808. doi: 10.1139/W07-048

Guimarães, R.A.; Pherez-Perrony, P.E.; Müller, H.; Berg, G.; Vasconcelos Medeiros, F.H.; Cernava, T. Microbiome-guided evaluation of Bacillus subtilis BIOUFLA2 application to reduce mycotoxins in maize kernels. Biol. Cont. 2020 150, 104370. doi: 10.1016/j.biocontrol.2020.104370

Johnston-Monje, D.; Raizada, M.N. Conservation and diversity of seed associated endophytes in Zea across boundaries of evolution, ethnography and ecology. PLOS ONE 2011 6, e20396. doi: 10.1371/journal.pone.0020396

Passera, A.; Follador, A.; Morandi, S.; Miotti, N.; Ghidoli, M.; Venturini, G.; Quaglino, F.; Brasca, M.; Casati, P.; Pilu, R.; Bulgarelli, D. Bacterial Communities in the Embryo of Maize Landraces: Relation with Susceptibility to Fusarium Ear Rot. Microorganisms 2021, 9, 2388. https://doi.org/10.3390/microorganisms9112388

Author Response

The present manuscript presents a study that starts from a strong hypothesis and works in a scientifically-sound way to investigate the proposed thematic.

The methods employed to investigate the microbiota are correct and coherent but, in the materials and methods section, giving some more explanation regarding the rationale behind the selection of the species to investigate would be helpful to the readers later on. Likewise, including some info, even just the list of species, in the main text and not only in the Supplementary Material would be appreciated.

Answer: We thank the reviewer’s comments here. A paragraph of the detailed selection of the plant species has been including in subsection 2.1 plant seed collection and surface sterilization, as below

“Due to diverse plant life features (herb, vine and shrub) with divergent seeds may harbor different seed-borne microbiota, we selected ten plants that have evolved low seed mass (‘light seeds’) with specific dispersal strategies, including two hydrochoric herbs (CS, Euryale ferox; PFC, Nuphar pumila), two anemochoric herbs (QM, Dianthus superbus; CJSZ, Dianthus repens), three anemochoric vines (DHQT, Illigera grandiflora; XNFCZ, Combretum griffithii; SFCZ, Combretum wallichii) and three tree species (QGLR, Terminalia myriocarpa; DLR, Terminalia franchetii; CZLR, Terminalia franchetii).”

The results section would need a great overhaul of the figures presented, which all lack in clarity that could help the readers.

Figure 1 is hard to compare due to the scale. Using a simple log10 scale on the Y-axis should, in my opinion, allow direct comparison of the quantities without relying on split panels and still not managing to understand what some bars represent (for example, SFCZ in panel A... it's below 4.00E7 but above 0... how much is it?).

Answer: As recommended by the reviewer, the Figure 1 has been reworked using log 10 scale on the Y-axis.

On all the figures, using different colors or marker shapes to identify different plant families and - most importantly - to allow the readers to immediately identify the anemochores and the hydrochores would make the figures much, much easier to interprete for a reader. As it is, I had to continuously go back and forth between figures and the first caption to make sense of the pictures and see what the results were saying.

Answer: We thank the reviewer’s comments. The Figure 2-Figure 5 have all been revised using clear legends to indicate plant families and their assigned dispersal strategies (anemochores/hydrochores). Moreover, to make readers job easier, we added detailed explanations for plant abbreviations in each figure caption.

Lastly, while the discussion section is appropriate, I see that the authors did not cite many works regarding the seed microbiota of cereals. Especially regarding the considerations of H1, that the plant-seed microbiota showed different, species- or even genotype-specific signatures, there are already indications of things going in that direction in the literature.

I therefore suggest one or more of the following articles for consideration in discussing the results and/or for the introduction:

Rijavec, T.; Lapanje, A.; Dermastia, M.; Rupnik, M. Isolation of bacterial endophytes from germinated maize kernels. Can. J. Microbiol. 2007 53, 802-808. doi: 10.1139/W07-048

Guimarães, R.A.; Pherez-Perrony, P.E.; Müller, H.; Berg, G.; Vasconcelos Medeiros, F.H.; Cernava, T. Microbiome-guided evaluation of Bacillus subtilis BIOUFLA2 application to reduce mycotoxins in maize kernels. Biol. Cont. 2020 150, 104370. doi: 10.1016/j.biocontrol.2020.104370

Johnston-Monje, D.; Raizada, M.N. Conservation and diversity of seed associated endophytes in Zea across boundaries of evolution, ethnography and ecology. PLOS ONE 2011 6, e20396. doi: 10.1371/journal.pone.0020396

Passera, A.; Follador, A.; Morandi, S.; Miotti, N.; Ghidoli, M.; Venturini, G.; Quaglino, F.; Brasca, M.; Casati, P.; Pilu, R.; Bulgarelli, D. Bacterial Communities in the Embryo of Maize Landraces: Relation with Susceptibility to Fusarium Ear Rot. Microorganisms 2021, 9, 2388. https://doi.org/10.3390/microorganisms9112388

Answer: We thank the reviewer’s valuable inputs. The suggested four papers have been carefully read and included in the main text.

In the introduction part, we included “David studied the conservation and diversity of seed associated endophytes in Zea, found that major seed bacterial microbiota in the Zea wild ancestor persist in diverse domesticated maize [13], and seed bacterial endophyte community composition changed with its host plant phylogeny [13]”.

In the discussion section, we wrote a new paragraph, as below:

“Although the ecological impacts of observed anemochore seeds have not been experimentally tested, most anemochore seed endophytes such as Methylobacteria, Pantoea and Pseudomonas, were also identified in other plants (maize cultivars). They exhibit the ability of stimulating plant growth by phosphate solubilization and pathogen antagonization [13]. An application of fungicides with the Bacillus subtilis strain was identified as a promising strategy to control Fusarium verticillioides-caused diseases [49]. Moreover, it is commonly accepted that seed endophytes could spread systemically through the seeding phyllosphere plant, exit the root and colonize the rhizosphere [13,21], playing vital roles in plant health, disease resistance and growth-promotion [13,50]. For anemochore, the extensive existence and co-evolution of these seed endophytes with their host plant, might be a key biotic factor for plant surviving in harsh environments.”